# Early alveolar macrophage response and IL-1R-dependent T cell priming determine transmissibility of *Mycobacterium tuberculosis* strains

Arianne Lovey [1], Sheetal Verma[1], Vaishnavi Kaipilyawar [1], Rodrigo Ribeiro-Rodrigues [2], Seema Husain[3], Moises Palaci[2], Reynaldo Dietze[2,4], Shuyi Ma[5,6,7,8], Robert D. Morrison[9], David. R. Sherman [10], Jerrold J. Ellner[1] & Padmini Salgame [1]✉

Mechanisms underlying variability in transmission of *Mycobacterium tuberculosis* strains remain undefined. By characterizing high and low transmission strains of *M.tuberculosis* in mice, we show here that high transmission *M.tuberculosis* strain induce rapid IL-1R-dependent alveolar macrophage migration from the alveolar space into the interstitium and that this action is key to subsequent temporal events of early dissemination of bacteria to the lymph nodes, Th1 priming, granulomatous response and bacterial control. In contrast, IL-1R-dependent alveolar macrophage migration and early dissemination of bacteria to lymph nodes is significantly impeded in infection with low transmission *M.tuberculosis* strain; these events promote the development of Th17 immunity, fostering neutrophilic inflammation and increased bacterial replication. Our results suggest that by inducing granulomas with the potential to develop into cavitary lesions that aids bacterial escape into the airways, high transmission *M.tuberculosis* strain is poised for greater transmissibility. These findings implicate bacterial heterogeneity as an important modifier of TB disease manifestations and transmission.

[1] Center for Emerging Pathogens, Department of Medicine, Rutgers-New Jersey Medical School, Newark, NJ, USA. [2] Núcleo de Doenças Infecciosas, NDI/Universidade Federal do Espirito Santo-UFES, Vitoria, Brazil. [3] The Genomics Center, Rutgers—New Jersey Medical School, Newark, NJ, USA. [4] Global Health & Tropical Medicine—Instituto de Higiene e Medicina Tropical—Universidade Nova de Lisboa, Lisbon, Portugal. [5] Center for Global Infectious Disease Research, Seattle Children's Research Institute, Seattle, WA, USA. [6] Division of Infectious Diseases, Department of Pediatrics, University of Washington, Seattle, WA, USA. [7] Pathobiology Program, Department of Global Health, University of Washington, Seattle, WA, USA. [8] Department of Chemical Engineering, University of Washington, Seattle, WA, USA. [9] Laboratory of Malaria Immunology and Vaccinology, National Institute of Allergy and Infectious Diseases, NIH, Bethesda, MD, USA. [10] Department of Microbiology, University of Washington, Seattle, WA 98109-8070, USA. ✉email: padmini.salgame@rutgers.edu

Halting transmission of *Mycobacterium tuberculosis* (Mtb) is essential to reduce the burden of tuberculosis (TB) globally and to achieve the goals of the End TB Strategy[1]. Transmission of Mtb from an individual with tuberculosis to others is significantly influenced by the amount of infectious airborne droplets expelled from an infected individual by expiratory processes such as coughing and sneezing[2]. Air sampling studies conducted in TB wards found unequal transmission of infection from the patients to test animals despite comparable sputum positivity[3–5]. Studies with cough aerosol measuring systems[6,7] and molecular epidemiology studies[8–10] also indicate that some individuals with TB are more infectious than others. However, the mechanisms responsible for this variability in transmission between individuals with TB are not well understood.

*Mycobacterium tuberculosis* Complex (MTBC) has diverged into several human-adapted lineages[11] and accumulating evidence indicates that genotypic diversity can determine the virulence and transmissibility of clinical MTBC isolates and influence the outcome of TB disease (reviewed in[11,12]). The "modern" Mtb lineages induce low levels of inflammatory cytokine[13–15] and are associated with faster growth rate in macrophages[16] compared to "ancient" Mtb lineages. Consistent with attenuated inflammatory response, sub-strains of the W-Beijing strain causes more severe disease in humans and in animal models[17–21]. However, comparing both "ancient" and "modern" lineages together revealed no direct correlation between low cytokine induction in vitro and high virulence in mice[16]. The studies described above used clustering versus non-clustering of Mtb isolates characterized by molecular genotyping as a proxy for transmission and hence, have not considered environmental factors and specifically the infectiousness of the index case. To address these limitations, we conducted a study in Brazil of household contacts (HHC) of infectious TB cases. We found heterogeneity of Mtb transmission within households not readily explained by household contact or environmental factors[22]. However, the study found that the index case of high transmission households had significant association with stronger cough and a trend towards significance for the presence of cavitation in chest radiograph[22]. Given that numerous bacilli are found at the surface of cavities[23] with increased potential for escape into the airways suggests that strain-induced pathology-regulating factors in the infected individuals could be a significant contributor to the variability in transmission.

To dissect the complex interaction of the pathogen with the host leading to heterogeneity in transmission, we examined, in a previous study, Mtb strains from the source index case of high transmission (HT) and low transmission (LT) households in the C3HeB/FeJ mouse model of TB[24]. We found that C3HeB/FeJ mice infected with Mtb-LT strains exhibited significantly higher bacterial burden compared to Mtb-HT strains and developed diffused neutrophilic inflammatory lung pathology. In stark contrast, mice infected with Mtb-HT strains developed discrete caseating granulomas, a lesion type with high potential to cavitate and none developed diffused inflammatory pathology[24]. Overall, the findings from our published study indicate that by inducing rapid progressive pathology, Mtb-LT creates an environment that is not conducive for the escape of bacteria into the airways but one that leads to increased mortality, if not treated. Mtb-HT, on the other hand, causes minimal pathological disease and is contained within granulomas initially, but these granulomas subsequently progress to a caseating necrotic stage, creating an environment in which bacterial escape into the airways, and hence transmission, is possible. Studies in the C3HeB/FeJ model thus links clinical phenotype of Mtb strains with distinct experimental phenotype providing a powerful tool to further investigate the early host-pathogen interaction events. Of note,

both Mtb-HT and Mtb-LT strains belong to lineage 4, and yet exhibit strain-specific virulence traits, consistent with whole genome level study showing that even closely related strains show genetic differences[25].

Here, we report that differences in the early interaction of Mtb-HT and Mtb-LT with alveolar macrophages (AMs) set the stage for the differential trajectory in bacterial growth and granuloma formation. Uptake of Mtb-HT by AMs led to rapid localization to the lung interstitium in an IL-1R-dependent manner, followed by rapid dissemination of bacteria to draining lymph nodes, Th1 priming, and recruitment of T cells to the lungs. The outcome of these early mechanisms was the induction of a granulomatous response and growth restriction of Mtb-HT. In contrast, the interaction of Mtb-LT with AMs stalled their migration to the interstitium, thereby impeding transport of bacteria to lymph nodes (LNs) and Th1 priming resulting in a substantial delay in the arrival of antigen-specific T-cells to the lungs. The lack of Th1 priming skewed T cell differentiation to a dominant Th17 response allowing for neutrophilic infiltration and subsequent disseminated pulmonary pathology. We have previously reported the characterization of a panel of Mtb-HT and Mtb-LT strains[24]. In this study, Mtb-HT1 and Mtb-LT1 were chosen as the prototypical high and low transmission strains. Mtb-HT2 and Mtb-LT2 were also studied in select experiments.

## Results

**Early dissemination of bacteria is impeded in Mtb-LT infection.** As previously observed[24], significantly greater replication was observed in the lungs of Mtb-LT1 compared to Mtb-HT1 at weeks 3, 4 and 11 weeks following infection (Fig. 1a). Examining bacterial growth at the earlier 2-week time point revealed similar lung colony forming units (CFUs) between Mtb-HT1 and Mtb-LT1 infected mice (Fig. 1a), but the notable observation was the significantly higher bacterial burden in the lymph nodes (Fig. 1b) and spleen (Fig. 1c) of Mtb-HT1 when compared with Mtb-LT1 infected mice.

**Delayed Th1 priming in Mtb-LT and skewing instead to Th17 differentiation.** Previous studies have shown that Mtb dissemination to the lymph node must occur prior to the development of an adaptive immune response[26] and that the resultant CD4[+] T cell recruitment to the lungs is directly proportional to the number of viable bacteria in the lymph node[27]. Therefore, we next examined the degree of activation and cytokine expression of T cells in the draining lymph nodes of 2 and 4 week-infected mice. We found that both total number of cells (Fig. 2a) and the number of CD4[+]CD69[+] activated T cells (Fig. 2b) were significantly higher at week 2 in the lymph nodes of Mtb-HT1 infected mice in comparison to Mtb-LT1 infected mice. Intracytoplasmic staining revealed that there was significantly increased IFNγ (Fig. 2c) and IL-17 (Fig. 2e) expressing cells among the CD4[+]CD69[+] T cells in Mtb-HT1 infected mice. This indicated that there was enhanced CD4[+] T cell priming at week 2 post infection in Mtb-HT1 infection as the appearance of CD4[+]CD69[+] T cells is the first evidence of CD4[+] T cell priming in the lymph nodes during Mtb infection[27]. CD44 is also a marker of T cell activation, and similarly, significantly increased IFNγ[+] (Fig. 2d) and IL-17[+] (Fig. 2f) cells were present among CD4[+]CD44[+] T cells from Mtb-HT1 infected mice at week 2 post infection. At week 4 post infection, activated CD4[+]CD69[+] T cell numbers increased in both groups and the numbers were now equivalent at this timepoint (Fig. 2b). However, the total number of IFNγ-secreting cells among the activated T cells in 4-week lymph nodes of Mtb-LT infected mice did not reach the level of Mtb-HT1 infected mice (Fig. 2c, d). Although IFNγ expressing

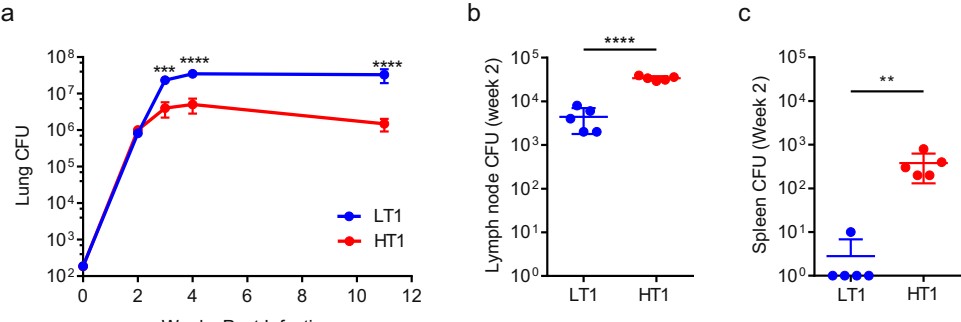

**Fig. 1 Increased dissemination of Mtb to lymph node and spleen in Mtb-HT1 infected mice.** C3HeB/FeJ mice were infected with ~100 CFU of the indicated strain via Glas-Col aerosol exposure. At the indicated time following infection, mice were sacrificed and CFU was determined in lung (**a**), draining lymph nodes (**b**) and spleen (**c**). Sample size of $n = 5$ mice were included for each strain and for each time point. Data are presented as mean values $+/-$ SD. Significance was determined using two-way ANOVA with Bonferroni post-test (**a**) and unpaired t tests (**b** and **c**). ** $P = 0.0095$, ****$P < 0.0001$. The data are representative of one of two individual experiments with Mtb-HT1 and Mtb-LT1 and was repeated with Mtb-HT2 and Mtb-LT2. Source data are provided as a Source Data file.

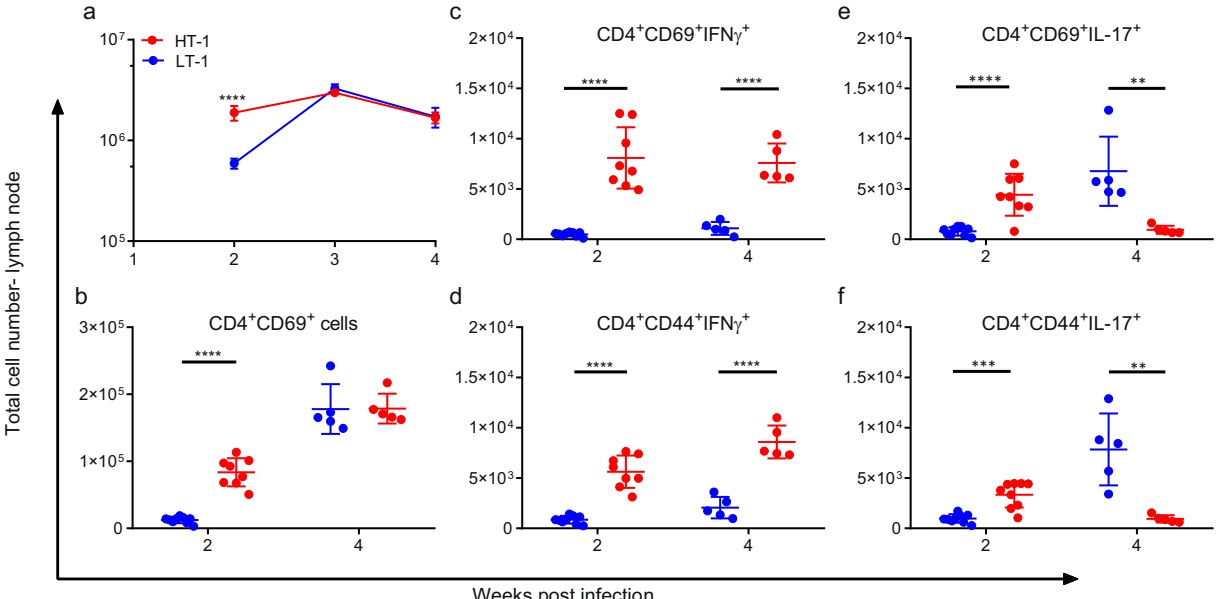

**Fig. 2 Increased dissemination in conjunction with early Th1 priming in the lymph nodes of Mtb-HT infected mice.** C3HeB/FeJ mice were infected with ~100 CFU of the indicated strain via Glas-Col aerosol exposure. At the indicated time post infection mice were sacrificed and mediastinal lymph nodes were isolated. Single cells were stimulated for 3 h with Mtb-H37Rv lysate and mycobacterial ESAT-6 and CFP-10 peptide pools and stained with a panel of cell surface antibodies, fixed and permeabilized, and then stained with a panel of intracellular antibodies. Cellular recruitment to the lymph node presented as absolute number (**a**) and percent of activated CD4+ T cells in the lymph nodes (**b**) was evaluated in Mtb-LT1 and Mtb-HT1 infected mice at weeks 2, 3, and 4 post infection. Percent of CD69+ (**c**) and CD44+ (**d**) CD4+ T cells expressing IFNγ+ and percent of CD69+ (**e**) and CD44+ (**f**) CD4+ T cells expressing IL-17 evaluated in the lymph nodes at weeks 2 and 4 post infection. Data are presented as mean values $+/-$ SD. Source data are provided as a Source Data file. Sample size of $n = 8$ mice were included for 2-week time point and $n = 5$ mice for week 3 and 4 time points. Significance was determined using two-way ANOVA with Bonferroni post-test (**a**) and unpaired t test (**b**–**f**). **$P = 0.0026$ in panel (**e**), **$P = 0.0056$ in panel (**f**), ***$p < 0.001$, ****$p < 0.0001$. The experiment was repeated with Mtb-HT2 and Mtb-LT2.

cells were maintained in Mtb-HT1 at 4 weeks of infection, there was nonetheless a marked contraction of IL-17$^+$ T cells (Fig. 2e, f). In stark contrast, in Mtb-LT1 infection, IL-17 expressing CD4$^+$ T cells increased and were significantly higher in comparison to Mtb-HT1 infection (Fig. 2e, f). Together, the data so far indicate that there is not only a delay, but also a phenotypic difference in the T cell response in Mtb-LT1 infected mice compared to Mtb-HT1 infected mice. This finding establishes that in Mtb-LT infection, the delayed Mtb dissemination to the lymph nodes does not result in complete suppression of T cell activation and differentiation, but instead it is later skewed towards Th17 dominant response.

**Early establishment of distinct immune cell environment in the lungs of Mtb-HT and Mtb-LT infected mice.** To address whether differences in T cell priming may have been influenced by early changes in lung microenvironment, we analyzed the cellular infiltrates in the lungs of Mtb-LT1 and Mtb-HT1 infected mice at 2 weeks following infection. We found that the total number of cells recruited to the lungs of Mtb-HT1 infected mice was higher than Mtb-LT infected mice at week 2 post infection (Fig. 3a) and that the percent and the total number of neutrophils, recruited macrophages, inflammatory monocytes including CCR2$^+$ and CD301a$^+$ inflammatory monocytes were similar in Mtb-LT1 and Mtb-HT1 infections (Supplementary Figs. 1 and 2).

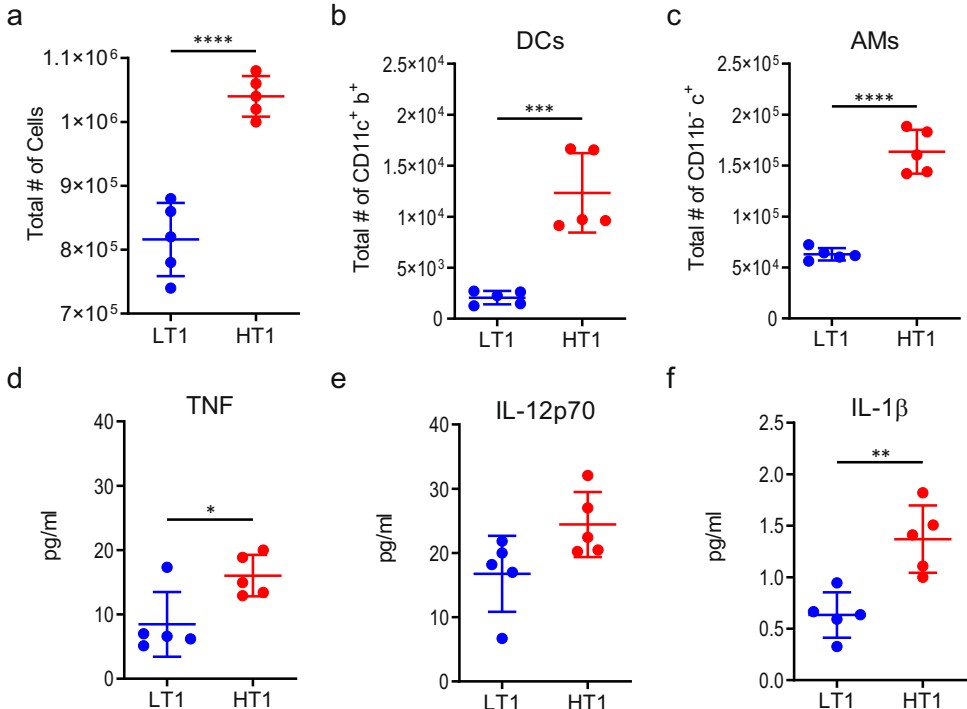

**Fig. 3 Different microenvironments present in the lungs of Mtb-HT and Mtb-LT infected mice at week 2 post infection. a–f** C3HeB/FeJ mice were infected with ~100 CFU of the indicated strain via Glas-Col aerosol exposure. At the indicated time post infection mice were sacrificed and single cell suspensions and total number of recruited cells (**a**) cDCs (CD3-CD19-CD11c+MHCII+) (**b**) and AMs (CD3-CD19-CD11b-CD11c+SiglecF+ (**c**) were calculated following acquisition on LSRFortessa X-20 and FlowJo analysis. MSD ELISA of lung lysates was used to determine TNFα (**d**), IL-12 (**e**), and IL-1β (**f**) protein levels. Data are presented as mean values +/− SD. Source data are provided as a Source Data file. Sample size of $n = 5$ mice was included in each group. Significance was determined using unpaired t test. *$P = 0.0217$, **$P = 0.0031$, ***$P < 0.001$, ****$P < 0.0001$. The experiment was repeated with Mtb-HT2 and Mtb-LT2.

Interestingly, the total number of conventional dendritic cells (cDCs) (CD3-CD19-CD11C+MHCII+) and AMs (CD3-CD19-CD11b-CD11c+SiglecF+) were increased in Mtb-HT1 infected mice compared to Mtb-LT1 infected mice (Fig. 3b, c; Supplementary Fig. 1). These data indicate that the AM population undergoes expansion in Mtb-HT1 infection, consistent with the proliferation of AMs in the airway and interstitium as observed by others[28]. The difference in cellular recruitment was accompanied by increased expression of TNF and IL-1β in Mtb-HT1-infected lungs (Fig. 3d, f) but not IL-12 (Fig. 3e).

Next, we determined whether the difference in T cell priming in the lymph nodes affected antigen-specific CD4+ T cell recruitment to the lung. Therefore, we first examined the kinetics of T cell recruitment to the lungs of Mtb-LT1 and Mtb-HT1 infected mice. We found that at week 2 post infection, consistent with rapid priming in the lymph nodes, there was an increase in the total number of CD4+ T cells in the lungs of Mtb-HT1 infected mice compared to Mtb-LT1 (Supplementary Fig. 3). However, by week 4, in line with the increased bacterial replication seen at this time point, total number of CD4+ T cells in Mtb-LT1 infected mice rose significantly and was higher than that in Mtb-HT1 infection (Supplementary Fig. 3). To further determine whether T cell recruitment resulted in changes to the lung cytokine milieu, we examined IFNγ and IL-17 levels in lung lysates obtained at weeks 2, 3, and 4 post infection. IFNγ was detected as early as week 2 of infection in Mtb-HT1 infected mice but remained undetectable up to 3 weeks in Mtb-LT1 infected mice, and then rising to levels equivalent to Mtb-HT1 at 4 weeks (Fig. 4a). IL-17A was below the limit of detection in both Mtb-LT1 and Mtb-HT1 infected mice at week 2 post infection, however, at week 3 and 4 post infection IL-17A levels

were greater in Mtb-LT1 infected mice compared to Mtb-HT1 infected mice (Fig. 4b).

Consistent with the levels of secreted protein, frequency of IFNγ expressing CD4+ T cells was higher in the lungs of Mtb-HT infected mice (Fig. 4c), and reciprocally IL-17 expressing T cells increased in frequency in Mtb-LT infected mice (Fig. 4d). To examine if Mtb-LT infection would inhibit the Th1 response of Mtb-HT infected mice, we performed a mixed experiment where mice were dually infected with a low dose of both Mtb-HT1 and Mtb-LT1. We found that at week 4, the percent of CD4+IFNγ+ T cells in the lungs of mice receiving Mtb-HT1 alone and those receiving the dual infection was comparable and higher to that present in Mtb-LT infected mice (Fig. 4e).

Collectively, these data demonstrate that the differential T cell priming in the lymph nodes of Mtb-HT1 and Mtb-LT1 infected mice translates into the development of distinct lung immune environment, with early IFNγ response in the lungs of Mtb-HT1 infected mice and a Th17 dominant response in Mtb-LT1 infected mice. Furthermore, Mtb-LT1 does not suppress the early immune activation induced by Mtb-HT1.

**T cells from Mtb-HT infected mice confer control of bacterial replication in Mtb-LT infection.** Following infection, Mtb continues to replicate in the lungs of mice until control is achieved by the arrival of Th1 effector cells. The data presented so far indicate that the initial delay in recruitment of Th1 cells to Mtb-LT1 infected lung allows Mtb-LT1 to continue to replicate until Th1 cells ultimately arrive and the bacterial burden stabilizes, albeit at a significantly higher setpoint. Therefore, we asked if adoptive transfer of T cells from Mtb-HT1 infected mice will

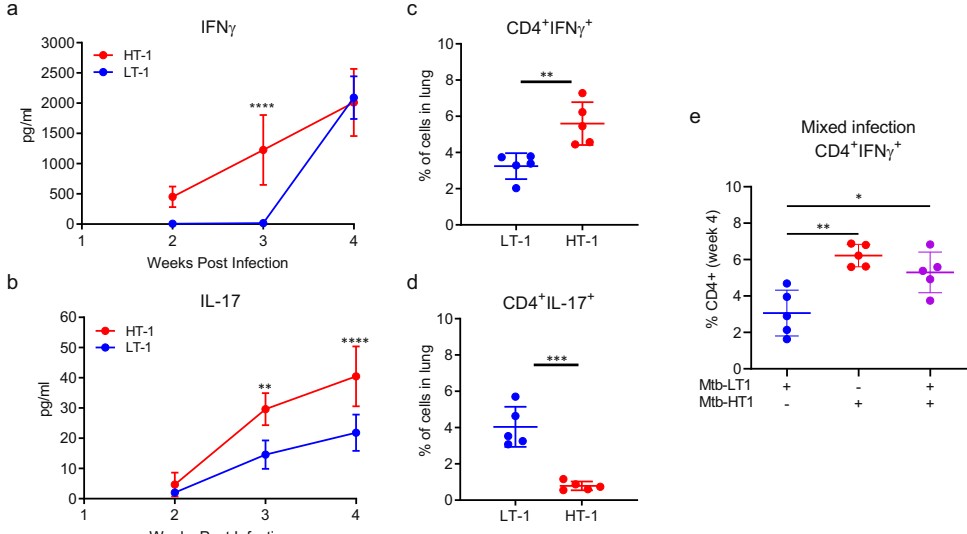

**Fig. 4 T cell response is skewed towards Th17 in Mtb-LT infection. C3HeB/FeJ mice were infected with ~100 CFU of the indicated strain via Glas-Col aerosol exposure.** IFNγ (**a**) and IL-17A (**b**) protein expression at weeks 2, 3, and 4 post infection was determined by ELISA. Single cell suspensions prepared from the lungs were stimulated for 3 h with Mtb-H37Rv lysate and mycobacterial ESAT-6 and CFP-10 peptide pools and stained with a panel of cell surface antibodies, fixed and permeabilized, and then stained with a panel of intracellular antibodies. Data are presented as percent of IFNγ+CD4+ (**c**) or IL-17+CD4+ (**d**) T cells. In a separate experiment Mtb-HT1, Mtb-LT1, and dual Mtb-HT1/LT1 infected mice were evaluated for IFNγ T cells at week 4 post infection (**e**). Data are presented as mean values +/− SD. Source data are provided as a Source Data file. Sample size of $n = 5$ mice was included in each group. Significance was determined using two-way ANOVA with Bonferroni post-test (**a** and **b**), unpaired t test (**c** and **d**) or one-way ANOVA with Tukey's correction (**e**) *$P = 0.0134$ (panel **e**), **$P = 0.0011$ (panel **b**), **$P = 0.0052$ (panel **c**) and $P = 0.0011$ (panel **e**), ***$P < 0.001$, ****$P < 0.0001$. The experiment was repeated with Mtb-HT2 and Mtb-LT2.

endow Mtb-LT1 infected mice the ability to better control bacterial replication. To address this hypothesis, CD3+ T cells were pan-purified from the lymph nodes and spleen of Mtb-HT1 (HT-CD3) and Mtb-LT1 (LT-CD3) infected mice at 6 weeks of infection. $2 \times 10^6$ each of HT-CD3 T cells or LT-CD3 T cells were adoptively transferred into groups of mice. The control group did not receive any T cells. Half of the mice in each group were infected with Mtb-HT1 and the other half with Mtb-LT1. Bacterial burden in the lungs was then determined at 11 weeks post-infection. As shown in Fig. 5a, we found that Mtb-LT1 infected mice adoptively transferred with HT-CD3 T cells had a significant decrease in bacterial burden compared to control mice whereas a similar decrease was not observed when LT-CD3 T cells were transferred. Adoptive transfer of HT-CD3 T cells or LT-CD3 T cells had no effect on the lung CFU of Mtb-HT1 infected mice. The reduction in bacterial numbers in the lungs of Mtb-LT1 infected mice receiving HT-CD3 T cells was accompanied by a significant reduction in protein levels of IFNγ, IL-17, and TNF (Fig. 5b–d) whereas IL-1β level was not significantly reduced (Fig. 5e). As expected, in conjunction with lower bacterial burdens, all three groups of Mtb-HT1 infected mice had diminished proinflammatory cytokine expression compared to Mtb-LT1 infected mice (Fig. 5b–e).

**IL-1R-dependent rapid bacterial dissemination to LNs and Th1 priming mediates bacterial control in Mtb-HT1 infection and prevents inflammatory neutrophilic pathology.** The data thus far indicate that variance in the rate of Mtb dissemination into LNs and T cell priming segregates Mtb-HT1 and Mtb-LT1 infections into Th1 and Th17 dominant immune responses, respectively. We then addressed the mechanism controlling the divergence in the rate of bacterial dissemination in the two infection groups. The first cells to take up inhaled Mtb are the alveolar macrophages (AM) present in the alveolar space[28],

therefore, we looked at early gene expression pattern in airway AMs from Mtb-HT1 and Mtb-LT1 infected mice. Mice were intratracheally instilled with anti-mouse SiglecF-PE-CF594 antibody prior to sacrifice to stain and sort airway-resident AMs. At D13 post infection, we sorted airway-resident (defined as CD11b-CD11c+SiglecF+) from whole lung single cell suspensions of Mtb-HT1 and Mtb-LT1 infected mice. RNA was extracted from these AMs and used to quantify the expression of 254 inflammatory genes (Mouse Inflammation V2 Panel, NanoString). Differential gene expression analysis demonstrated that AMs from Mtb-HT1 infected mice significantly upregulated several pro-inflammatory cytokine genes including *TNF, IL1A, IL1B,* and *HMGB1,* inflammasome *NLRP3* and leukocyte and granulocyte chemo-attractants *CCL4, CCL2, CXCL2, CXCL3, CCL3,* and *CXCL1* (Fig. 6a). In conjunction with increased pro-inflammatory mediators, inflammation regulatory genes *IL1RN* and *TNFAIP3* were also upregulated by these AMs from Mtb-HT1 infected mice. In comparison, AMs from Mtb-LT1 infected mice only significantly upregulated *MAP3K1,* a mitogen-activated protein kinase involved in cellular signal transduction pathways. These data indicate that at this early timepoint following infection, AMs from Mtb-HT1 infected mice are preferentially activated to express a number of cytokines and chemokines, including IL-1α and IL-1β. AM migration to the interstitium is propelled by IL-1R signaling on non-hematopoietic cells[28], leading us to test whether the early difference in AM activation and IL-1R signaling in Mtb-HT and Mtb-LT infections influences migration and the subsequent differential bacterial dissemination and adaptive T cell response pattern in the two infections.

The intratracheal instillation of SiglecF enabled us to not only mark the airway-resident AMs as SiglecF-PE-CF594+ve, but to also distinguish them from those already migrated into the interstitium, where since they are unable to react with the antibody are negative for Siglec-PE-CF594−ve expression (Supplementary Fig. 4). Next, we quantified AM migration in

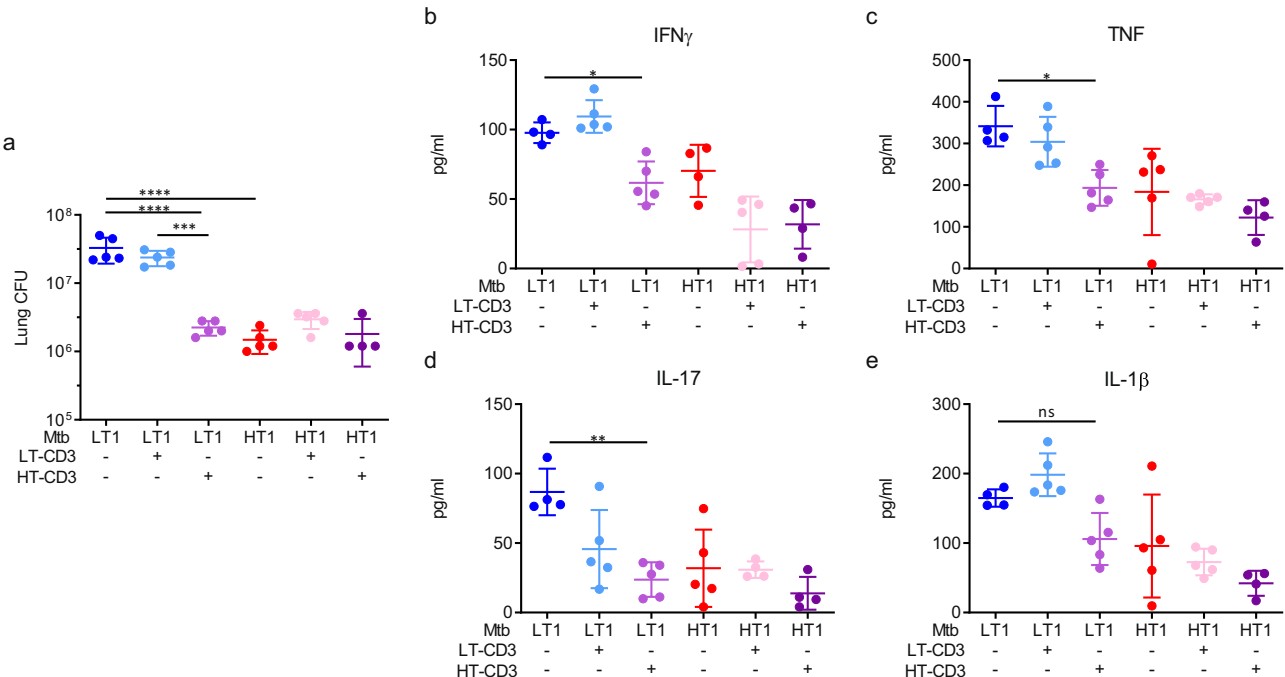

**Fig. 5 Effector T cells from Mtb-HT infected mice reduce bacterial burden and inflammation in Mtb-LT infected mice.** Three days prior to infection naïve mice were retro-orbitally injected with PBS or $2 \times 10^6$ CD3+ T cells isolated from 6 to 7-week Mtb-LT1 or Mtb-HT1 infected mice. On the day of infection, half of the mice in each group were infected with Mtb-HT1 and the other half with Mtb-LT1 via Glas-Col aerosol exposure. At 11 weeks post-infection, lung bacterial burden was determined and presented as CFU (**a**) and protein levels of IFNγ (**b**), TNF (**c**), IL-17 (**d**), and IL-1β (**e**) was measured in lung lysates by ELISA. Data are presented as mean values +/− SD. Source data are provided as a Source Data file. Sample size of $n = 5$ mice was included in each group. Significance was determined using one-way ANOVA with Tukey's correction *$P = 0.0421$ (IFNγ panel **b**), **$P = 0.0013$ (IL-17, panel **c**) *$P = 0.0132$ (TNF, panel **b**), ***$P < 0.001$, ****$P < 0.0001$. The experiment was repeated with Mtb-HT2 and Mtb-LT2.

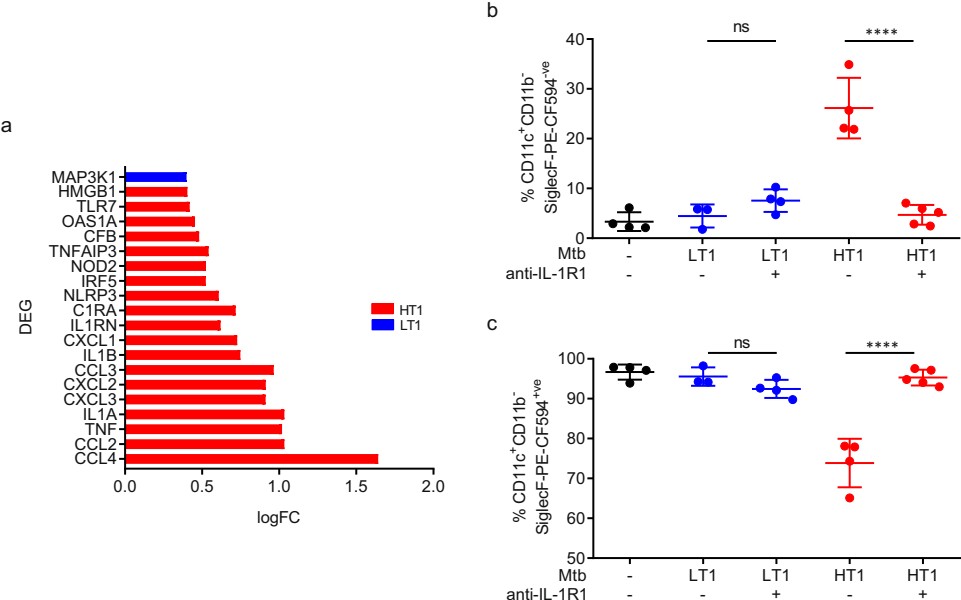

**Fig. 6 IL-1R dependent migration of alveolar macrophages in Mtb-HT infection. C3HeB/FeJ mice were infected with ~100 CFU of Mtb-LT1 or Mtb-HT1 via Glas-Col aerosol exposure.** At 13 days post infection, fluorescent-labeled anti-SiglecF antibody was intratracheally administered 30 min prior to sacrifice to stain and sort airway resident cells. Differential gene expression analysis by NanoString was performed on airway-resident AMs (CD11b-CD11c +SiglecF+) sorted from single cell lung suspensions pooled from five Mtb-HT1 and five Mtb-LT1 infected mice and the data are presented to show log fold change (logFC) of top DEG (Y-axis, $P < 0.05$) (**a**). For IL-1R dependent AM migration, groups of infected mice were administered intra-peritoneally on days 8, 10, and 12 post infection with either 200 μg α-IL-1R1 antibody or isotype control. Data are presented as airway-resident AMs (CD11b-CD11c+SiglecF+) (**b**) and AMs migrated into the interstitium (CD11b-CD11c+SiglecF−) (**c**). Data are presented as mean values +/− SD (panels b and c). Source data are provided as a Source Data file. Sample size of $n = 4$ mice were included for the first 4 groups and sample size of $n = 5$ mice for group 5 for panels (**b, c**). Significance was determined using one-way ANOVA with Tukey's correction. ****$P < 0.0001$. The data are representative of one of two individual experiments with Mtb-HT1 and Mtb-LT1 and was repeated with Mtb-HT2 and Mtb-LT2.

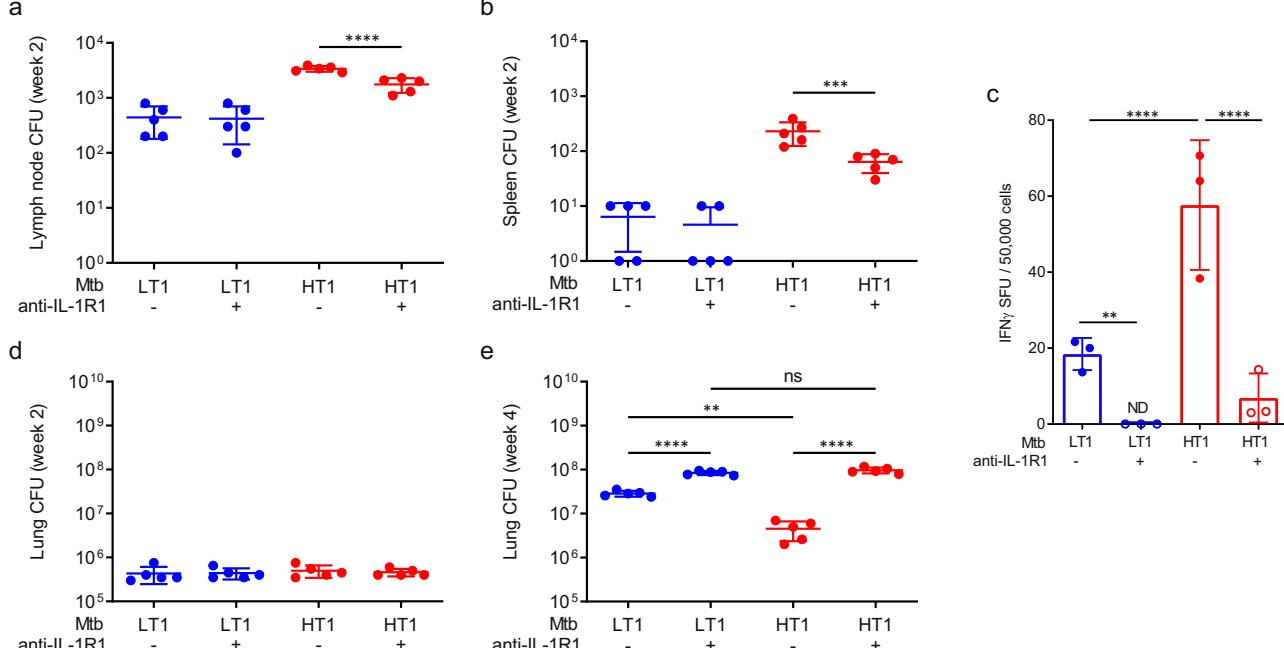

**Fig. 7 Rapid T cell priming and control of bacterial replication in Mtb-HT infection is dependent on IL-1R1 signaling.** C3HeB/FeJ mice were infected with ~100 CFU of the indicated strain via Glas-Col aerosol exposure. On days 8, 10, and 12 post infection 200μg of α-IL-1R1 antibody (+) or isotype control (−) was administered to the mice. Bacterial burden was determined at 2 weeks in the mediastinal lymph nodes (**a**), spleen (**b**), and lung (**d**) and at 4 weeks in the lung (**e**). At 2 weeks of infection, the frequency of IFNγ-secreting T cells was measured by ELISPOT, and data are presented as spot forming units (SFU)/50,000 LN cells. Data are presented as mean values +/− SD. Source data are provided as a Source Data file. Sample size of $n = 5$ mice were included for panels (**a**, **b**, **d**, **e**) and sample size of $n = 3$ mice was included for panel (**c**). Significance was determined using one-way ANOVA with Tukey's correction. **$P = 0.0064$ (panel **c**), **$P = 0.0032$ (panel **e**), **$P < 0.01$, ***$P < 0.001$, ****$P < 0.0001$. The data are representative of one of two individual experiments with Mtb-HT1 and Mtb-LT1.

Mtb-HT1 and Mtb-LT1 infected mice. In uninfected mice, we did not see any SiglecF PE-CF594$^{-ve}$ AMs (Fig. 6b); all cells stained positively for SiglecF-PE-CF594 (Fig. 6c), indicating that in these mice the AMs were mainly in the alveolar space. In Mtb-HT1 infected mice, we found that approximately 25% of AMs were SiglecF-PE-CF594$^{-ve}$ (Fig. 6b), indicating they had migrated out of the airway and into the interstitium by day 13 post infection. In contrast, very few AMs in Mtb-LT1 infected mice were SiglecF PE-CF594$^{-ve}$ (Fig. 6b) and instead, were SiglecF PE-CF594$^{+ve}$ (Fig. 6b), indicating that they were still in the alveolar space. Furthermore, in Mtb-HT1 infection, α-IL-1R1 treatment significantly reduced migration into the interstitium (Fig. 6b). No further delay in migration was observed in the Mtb-LT1 infection with α-IL-1R1 treatment (Fig. 6b, c).

Next, we determined whether the IL-1R dependent AM migration in Mtb-HT1 was an essential upstream event leading to bacterial dissemination to mediastinal LNs and the ensuant priming of Th1 cells. At 2 weeks of infection, as noted previously, there was increased CFU in LN (Fig. 7a) and spleen (Fig. 7b) of Mtb-HT1 compared to Mtb-LT1, whereas, in conjunction with inhibiting AM migration, transient IL-1R blockade also decreased Mtb dissemination to LNs and spleen in Mtb-HT1 infected mice (Fig. 7a, b). Remarkably, coincident with decreased dissemination following transient IL-1R blockade, the frequency of IFNγ-secreting T cells was also significantly reduced in the LNs of Mtb-HT1 infected mice at this time point compared to isotype-treated mice (Fig. 7c). As expected, Mtb-HT1 infected mice receiving isotype had significantly higher IFNγ secreting cells compared with Mtb-LT1 mice receiving similar treatment. The low frequency of IFNγ-secreting cells in Mtb-LT1 mice was completely extinguished by IL-1R blockade. Together, these data

indicate that, by controlling AM migration, IL-1R signaling affects the temporal activation of Th1 cells in the lymph nodes.

The deleterious effect of reduced IFNγ-secreting cells in LNs following IL-1R blockade was manifested later in the lungs. The early control of bacterial burden is similar between Mtb-LT and Mtb-HT infections at week 2 of infection (Fig. 7d). Importantly, IL-1R blockade did not affect Mtb replication at this time point in either Mtb-HT1 or Mtb-LT1 infections, but profoundly impacted bacterial control at week 4 post infection in Mtb-HT1 infected mice. Bacterial burdens in Mtb-HT1 infected mice in which IL-1R signaling had been transiently inhibited was increased 15-fold compared to isotype control, whereas bacterial burden in Mtb-LT1 infected mice, showed a small, albeit significant, increase following IL-1R blockade (Fig. 7e). As previously observed, Mtb-LT1 replication was significantly higher than Mtb-HT1 in the absence of IL-1R blockade. Stabilization of bacterial infection occurs around 3–4 weeks following infection when T cells recruited to the lung limit Mtb growth. By delaying Th1 activation, we find that Mtb-LT stalls the host's ability to restrict its growth.

The increased bacterial burden seen in Mtb-HT1 infected mice receiving α-IL-1R1 antibody was associated with increased recruitment of neutrophils and inflammatory monocytes (Fig. 8a, b). H&E staining revealed that Mtb-LT1 infected mice developed diffused inflammation that occupied much of the lung, while Mtb-HT1 infected mice exhibited cellular aggregates with significant area of the lung showing minimal cellular infiltration (Fig. 8c), akin to what we had seen previously[24]. However, Mtb-HT1 infected mice that received transient αIL-1R1 treatment developed necrotic pneumonia and histopathological disease like Mtb-LT1 (Fig. 8c), linking T cell priming and ensuant granulomatous response in the lung to IL-1R signaling.

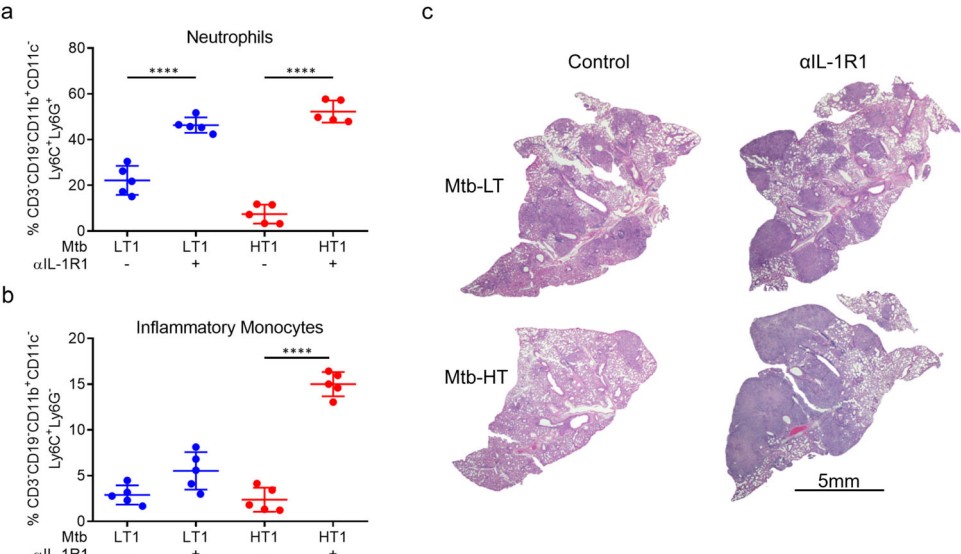

**Fig. 8 Transient inhibition of IL-1R signaling results in increased pathology in Mtb-HT infected mice.** C3HeB/FeJ mice were infected with ~100 CFU of the indicated strain via Glas-Col aerosol exposure. On days 8, 10, and 12 post infection 200ug α-IL-1R1 (+) or isotype control (−) was administered to the mice. At 4 weeks post infection mice were sacrificed and the right middle and inferior lung lobes were isolated to obtain single cells. Following perfusion, the left lobe was formalin-fixed, paraffin-embedded, and hematoxylin and eosin staining were carried out. Data are presented as percent of neutrophils (**a**) and inflammatory monocytes (**b**). Data are presented as mean values +/− SD. Source data are provided as a Source Data file. Representative H&E-stained lung sections for the indicated strains at week 4 post infection with and without transient IL-1R signaling. Scale bars: 5 mm (**c**). Five mice were included in each group. Significance was determined using one-way ANOVA with Tukey's correction. ****$P < 0.0001$.

**Sequence-based sublineage classification of Mtb-HT1 and Mtb-LT1.** Finally, we performed whole genome sequencing (WGS) for sublineage classification of Mtb-HT1 and Mtb-LT1. Our analysis pipeline assigned the Mtb-HT1 strain to sublineage 4.1.2.1, with an edit distance of 1 (FDR = 0.0009); and assigned strain Mtb-LT1 to sublineage 4.3.4.2, with an edit distance of 2 (FDR = 0.0014).

## Discussion

In this study, we identified the mechanism driving the divergence in pathogen growth and pathology in Mtb-HT and Mtb-LT infected C3HeB/FeJ mice. We found that Mtb-HT infected AMs, in an IL-1R-dependent manner, rapidly migrate into the interstitium to initiate Th1 priming and facilitate granuloma formation. In contrast, in Mtb-LT infection, Th1 activation is reined in because of the delayed migration of infected AMs resulting in a skewed Th17 response and inflammatory pathology in the lung.

The lung is constantly exposed to innocuous antigens and to maintain tissue immune homeostasis, AMs, in contrast to other macrophage subsets, have acquired a regulatory phenotype. Indeed, chromatin modifications in AMs are distinct from other tissue-resident macrophage populations[29]. To limit airway inflammation, AMs acquire an immunosuppressive and immunoregulatory phenotype[30] and express negative regulators such as CD200 receptor[31]. AMs also maintain an M2 phenotype with metabolism skewed to fatty acid oxidation[32]. During early in vivo infection in mice, AMs upregulate a *Nrf*-dependent transcriptional program that limits IL-1β and TNF production to impair the control of Mtb[33]. Overall, AMs present a conducive environment for the growth of Mtb and the threshold of immune regulatory environment must be overcome for an immune response to ensue following infection with Mtb. Our data showing differences in the early innate response, particularly IL-1α and IL-1β induction by AMs, in Mtb-HT and Mtb-LT infected mice suggest that AMs in Mtb-LT infection are inept at overriding their immunoregulatory environment.

The precise processes of how Th1 cells orchestrate the granulomatous response and control Mtb infection within this structure remain unclear, but timely activation and recruitment to the lungs of Th1 cells is a prerequisite to foster these processes. Mtb is detected earlier in the LNs of the relatively resistant C57BL/6 mice compared to the relatively susceptible C3H/HeJ strain and this correlates with the ability of C57BL/6 mice to better control Mtb infection[26]. Dendritic cell (DC) migration is essential to T cell priming[34,35] and there is evidence to suggest that Mtb may delay activation of effector T cells by inhibiting macrophage apoptosis[36–38], a step necessary for DC cross-priming[39,40]. In this study, we find that the delayed AM migration at the early stage of Mtb-LT infection impinges on the later adaptive phase of T cell priming. The further significant delay has major consequences to the type of antigen-specific effector T cells that develop in Mtb-LT infection. We found that the significant delay in Th1 cell activation in Mtb-LT infection deviates T cell differentiation to Th17 dominant response. In a mouse model of lymphocytic choriomeningitis virus (LCMV) and Mtb coinfection, susceptibility of the mice to Mtb infection is enhanced[41]. The pre-existing viral infection in the coinfected animals resulted in delayed Mtb-specific T cell priming and the diminished accumulation of antigen-specific Th1 cells. However, like Mtb-LT infected mice, there was instead an expansion of Th17 cells in the coinfected animals[41]. Although the cause of the delayed T cell priming in the coinfected mice and in Mtb-LT infected mice is different, it is interesting that a similar skewed Th17 cell expansion resulted in both infection models following delayed T cell priming. Mice lacking IFNγR in the nonhematopoietic compartment develop a dysregulated neutrophilic inflammatory pathology marked by over-expression of IL-17[42]. IFNγR signaling restricts IL-17 expansion through the induction of indoleamine-2,3-dioxygenase (IDO) and in the absence of the receptor this break on IL-17 expansion is removed[42]. It is conceivable that the reduced IFNγ expression observed in Mtb-LT1 infected mice and the consequent restricted signaling from IFNγR could have facilitated Th17 cell expansion in the mice.

IL-1 has been linked to increased pulmonary inflammation and TB disease severity. In susceptible mice, persistent IL-1β signaling increases granulocytic inflammation leading to increased immunopathology and bacterial replication[43]. In the relatively resistant C57BL/6 mouse strain, the onset of adaptive immunity and IFNγ production curtails excessive IL-1β production via NOS2-mediated inhibition of pro-IL1β processing[44]. A naturally occurring polymorphism in the human IL1B promoter region that leads to high IL-1β production is associated with the development of active tuberculosis, disease severity, and poor treatment outcome[45]. Consistent with its role in exacerbating pulmonary pathology, total lung inflammation was significantly reduced in Mtb-infected macaques treated with Anakinra, an IL-1 receptor antagonist, and the anti-TB drug Linezolid compared to animals receiving drug alone[46]. Although excessive IL-1β is immunopathological, availability of the cytokine during the early phase of infection is critical for host protection. Without intact IL-1 signaling such as that observed in mice lacking intact IL-1α, IL-1β, or IL-1R1, Mtb infection results in significant increase in bacterial burden and an increased number of Mtb-infected cells[47–51]. Several mechanisms have been proposed for IL-1R signaling mediated control of Mtb infection. For example, IL-1 suppresses the production of type-1 IFN that supports Mtb growth[52–55]. Another study revealed that Ly6G[hi] myeloid cells recruited to Mtb-infected lungs require IL-1R-dependent licensing to upregulate TNF-RI and produce reactive oxygen species for pathogen control[56]. The findings from our study implicate that by controlling AM migration, IL-1R signaling indirectly affects the rapidity of Mtb dissemination to the LNs for T cell priming and the succeeding temporal recruitment of T cells to the lungs and granuloma formation that is vital for pathogen growth restriction. Importantly, our data indicate that Mtb-LT can manipulate this pathway to delay Th1 priming and deviate granulomatous response to that of uncontrolled inflammation. Although at first, it seems contradictory to our finding that mice lacking the ligands for IL-1R (Il1a[−/−]Il1b[−/−] mice) develop Th1 adaptive immunity[56], it must, however, be noted that this study did not evaluate T cell priming and early IFNγ expression in the lymph nodes. As observed by us too with Mtb-HT and Mtb-LT infections, the similar IFN expression between WT and Il1a[−/−]Il1b[−/−] mice was seen at 30 days post-infection, most likely a reflection of the increased bacterial burden. In our study, by transiently inhibiting IL-1R signaling only during early infection, we were able to uncover a unique role for IL-1R signaling in initiating adaptive T cell response and granuloma formation and establish how Mtb strains can exploit this pathway to produce distinct disease outcomes in the host. Future studies should examine the role of IL-1R dependent early sequential mechanisms in dictating the long-term disease outcomes in Mtb exposed individuals and how that affects the transmission of the pathogen from the source case.

The heterogeneity in host response induced by Mtb-HT and Mtb-LT infections could also impact TB treatment outcomes. Despite successful treatment completion, a significant number of patients show pulmonary impairment[57–60]. Our findings would suggest that the exuberant inflammatory pathology induced by Mtb-LT is a risk factor for post TB lung damage. Future studies should characterize the impact of Mtb strain heterogeneity on immune mechanisms and treatment outcomes, enabling the design of novel effective host-directed therapeutic interventions.

MTBC Lineage 4 comprises 10 separate sublineages and by WGS we identified Mtb-HT1 as L4.1.2.1 and Mtb-LT1 as L4.3.4.2. Studies examining biological consequences of genotypic variations within sublineages is limited. In a study conducted in San Francisco, USA, a sub-lineage of Lineage 4 significantly differed in its transmissibility in comparison to other strains in the lineage[61]. A systematic review of MTBC in Africa revealed no link between Mtb lineage and clinical presentation of disease, but within Lineage 4 some sublineages were found to be associated more frequently with lymphnode TB than PTB[62]. What genotypic variations within the Mtb-HT and Mtb-LT sublineages regulate the distinct phenotypic outcome in the host remains to be determined. Potential targets include the highly polymorphic surface-expressed PE_PGRS33 protein with immunomodulatory properties[63–66]. Analysis of a large panel of Mtb clinical isolates revealed that those with altered PE_PGRS33 were more significantly coupled with the absence of cavitary disease suggesting a role in pathogenesis for this protein[63]. Because PE_PGRS33 is also polymorphic within a lineage[63], it is possible that variation in PE_PGRS33 expression may contribute to different immune phenotypes at the sublineage level with Mtb-HT1 and Mtb-LT1. Lipids in the cell wall of mycobacteria play important biological roles in determining virulence and host pathogenesis. Variation in the content of sulfatides and other acylated trehalose derivatives between strains within Lineage 4 has been observed[67]. Given that Mtb mutants compromised in sulfatide biogenesis exhibit reduced virulence in the mouse model of Mtb infection[68,69] and that Sulfolipid-1 stimulates Nociceptive Neurons and triggers cough[70] suggests that upcoming work should quantify the production of SL-1 and other lipids from Mtb-HT and Mtb-LT and examine the link to early cytokine response in AMs and subsequent distinct pathological response in the host. In addition to genomic variants, epigenetic mechanisms such as DNA methylation regulate gene expression. Mtb Lineage-specific methylation patterns[71] and expression of distinct DNA methyltransferase in Lineage 2 and 4 have been reported[72]. Different methylation patterns within Lineage 4 have also been described[73] which supports that examining the methylome of Mtb-HT and Mtb-LT could provide novel functional differences between them and insights into strain-dependent host immune and pathogenic responses. In conclusion, future studies should fully define the genetic features of high and low transmission strains of Mtb that regulate their in vivo phenotype and pattern of clinical presentation.

## Methods

**Ethics statement**. All animal experiments described in this study conform to the Rutgers University Biomedical Health Sciences-Newark (RBHS) and Institutional Animal Care and Use Committee (IACUC) Guidelines as well as NIH and USDA policies on the care and use of animals in research and teaching. Efforts were taken to ensure minimal animal pain and suffering and when applicable, approved anesthesia methods were employed for the same.

**Mouse strains**. Female C3HeB/FeJ (strain # 000658) mice approximately 5–7-week of age were purchased from the Jackson Laboratory and housed under pathogen-free conditions at the biosafety level 3 (BSL3) facility at the Public Health Research Institute for the duration of the study. The mice were housed at ambient temperature (72° +/− 4°), humidity range 35–70, and 12 h light–dark cycle (7 am on and 7 pm off).

**Bacterial strains**. In a household contact study in Brazil, Mtb isolated from the sputum of index cases was characterized as high transmission (Mtb-HT) or low transmission (Mtb-LT) based on the number of TST positive household contacts to build a panel of Mtb-HT and Mtb-LT strains[22]. In this study, we used a previously studied prototypic strain pair Mtb-HT1 and Mtb-LT1[24] that had been passaged twice. For the present study, Mtb-HT1 and Mtb-LT1 were passaged once in vivo, isolated after two weeks, and grown for 5–7 days in vitro to prepare stocks. To prepare Mtb stocks strains were passaged in Difco Middlebrook 7H9 Broth (BD 271310) supplemented with 10% OADC Enrichment Media (BD 212351) and 0.05% Tween 80 (Fisher BP338500). After reaching an OD600 of about 0.80 the culture was diluted with glycerol (Sigma G5516-1L). Stocks were frozen at −80 °C.

**Whole genome sequencing and sublineage classification for Mtb clinical strains**. The WGS of Mtb-HT1 and Mtb-LT1 was performed, the reads were mapped to H37Rv (NC_000962.3) reference genome and variants (SNP's and Indels) were called in the two samples. Supplementary Table 1 shows a list of 34 of the 396 polymorphisms unique to either Mtb-LT1 or Mtb-HT1 that were

identified. The presence of numerous unique variants indicated genetic heterogeneity between the two strains.

We have developed a computational workflow to assign phylogenetic sublineages of Mtb clinical strains directly from RNA-seq or WGS datasets. We extend the phylogenetic characterization approach by Coll and colleagues[74], which had defined 62 sublineage designations based on a set of single nucleotide polymorphism (SNP) markers (one per sublineage). Using the SNP data from Coll and colleagues[74], we identified sequence motifs consisting of 645 SNPs spanning 515 conserved essential Mtb genes that uniquely discriminate between the aforementioned 62 sublineages, with an average of 10 signature SNPs per sublineage-specific motif. Our approach of incorporating more than a single SNP in our sublineage-specific motifs yields improved sublineage assignment robustness and allows the estimation of an edit distance from the 'ideal' sublineage motif and a false discovery rate (FDR) for the sublineage assignment associated with each individual sample by counting the number of cumulative random perturbations of the motif needed to alter the sublineage call. Our computational workflow for Mtb sublineage assignment is publicly available as part of the DuffyNGS R package https://github.com/robertdouglasmorrison/DuffyNGS by using function pipe.BarcodeMotif.

**Aerosol infection.** Female mice aged 6–10 weeks were infected using the Glas-Col Full Body Inhalation Exposure System. Briefly, the mice were exposed to nebulized bacteria for 40 min at an inoculum dose standardized to deliver ~100 CFU of bacteria. 20–24 h post aerosol infection, 3–5 mice were euthanized and whole lungs were removed, homogenized, and plated to determine the infectious dose for each experiment.

**Mouse necropsy.** At appropriate time intervals post aerosol infection mice were euthanized and lungs, spleen, and draining lymph nodes were harvested. The right superior lobe of the lung was used for determining bacterial burden and the post-caval lobe was reserved for tissue gene expression. Following perfusion with PBS (Corning 21-031-CV) the right middle lobe, right inferior lobe, and left lobe were used either for histological studies or preparation of single cells. Spleen was used to determine the bacterial burden and lymph nodes were used either to determine bacterial burden, reserved for tissue gene expression, or used for the preparation of single cells.

**Estimation of bacterial burden.** Lung, spleen, or lymph node tissue was homogenized in 1 ml of PBS with 0.05% Tween 80. Serial 10× dilutions of the resultant products were prepared in PBS with 0.05% Tween 80 and plated on 7H11 plates. CFUs were counted after incubation at 37 °C in a non-$CO_2$ incubator for 14–28 days.

**Single cell preparation and flow cytometry for analysis of lung cellular infiltrates.** Lung lobes were digested with 20 μg/ml of collagenase D at 37 °C for 30 min, and the reaction was stopped by the addition of 5 mM of EDTA. Digested lung tissue was passed through 40 μm nylon membrane filter. Spleen and lymph nodes were passed through a 40 μm nylon membrane filter directly. Red blood cells were lysed by treating with ACK lysing buffer (Quality Biological 118-156-101). Single cell suspensions were surface stained at a 1:50 dilution in FACS Buffer (PBS, 10% FBS, 0.1% NaN3 sodium azide) with anti-mouse CD3-APC (clone 17A2; BD Pharmingen), anti-mouse CD19-APC (clone 1D3, BD Pharmingen), anti-mouse CD11b-APC-Cy7 (clone M1/70; BD Pharmingen), anti-mouse CD11c-BUV395 (clone HL3; BD Horizon), anti-mouse Ly6G-AF700 (clone 1A8; BD Pharmingen), anti-mouse Ly6C-PECy7 (clone AL-21; BD Pharmingen), anti-mouse SiglecF-PE-CF594 (clone E50-2440; BD Horizon), anti-mouse MHC-II (clone M5/114.15.2; BD Horizon), anti-mouse CCR2 (clone SA203G11; Biolegend) and anti-mouse CD301a (clone LOM-8.7; Biolegend). Samples were acquired using the LSRFortessa X-20 and analyzed using FlowJo software.

**Flow cytometric staining for intracellular cytokines.** Lung and lymph nodes were examined for intracellular cytokine production. Single cells were obtained as described above. Samples were then stimulated with 10 μg/ml H37Rv cell lysate (BEI NR-14822), 5 μg/ml ESAT6 peptide array (BEI NR-34824), and 5 μg/ml CFP10 peptide array (BEI NR-34825) in RPMI media for 3 h in a 37 °C, 5% $CO_2$ humidified incubator. Stimulated cells were then washed, and surface stained with anti-mouse CD3-FITC (clone 17A2, BD Pharmingen), anti-mouse CD4-V450 (clone RM4-5, BD Horizon), and anti-mouse CD8-PE (clone RM4-5, BD Pharmingen). Subsequently, cells were fixed and permeabilized using BD Cytofix/Cytoperm solution kit (BD 554714) as per the manufacturer's instructions. Permeabilized cells were stained for intracellular cytokines using anti-mouse IL-17A-AF647 (clone TC11-18H10, BD Pharmingen) and IFNγ-PECy7 (clone XMG1.2, BD Pharmingen). All antibodies were used at a 1:50 dilution. Samples were acquired using the LSRFortessa X-20 and analyzed using FlowJo software (v10.6.1).

**Alveolar macrophage migration and IL-1R inhibition.** To inhibit IL-1R signaling, infected C3HeB/FeJ mice were administered either 200 μg anti-mouse IL-1R (Bio X Cell BE0256) or 200 μg polyclonal Armenian hamster IgG (Bio X Cell BE0091) via

intraperitoneal injection on days 8, 10, and 12 post infection. Airway-resident alveolar macrophages were labeled by intratracheally instilling into anesthetized mice 0.4 μg anti-mouse SiglecF-PE-CF594 antibody (clone E50-2440; BD Horizon) in 100 μL of PBS. The mice remained suspended for approximately 30 s after instillation. The mice were then allowed to recover for 30 min prior to sacrifice via cervical dislocation to avoid bleeding into the lungs. The middle right, inferior right, and left lobes were harvested and digested in 20 μg/ml of collagenase D at 37 °C for 30 min. The post-caval lobe was used for gene expression analysis and the superior right lobe was used for CFU analysis. Digested lung tissue was passed through 40 μm nylon membrane filter. Red blood cells were lysed by treating with ACK lysing buffer (Quality Biological 118-156-101). Single cell suspensions were surface stained with anti-mouse CD3-APC (clone 17A2; BD Pharmingen), anti-mouse CD19-APC (clone 1D3, BD Pharmingen), anti-mouse CD11b-APC-Cy7 (clone M1/70; BD Pharmingen), anti-mouse CD11c-BUV395 (clone HL3; BD Horizon). All antibodies were used at a 1:50 dilution. Samples were acquired using the LSRFortessa X-20 and analyzed using FlowJo software (v10.6.1). Migrating AMs were defined as CD11b-CD11c+SiglecF− while non-migrating AMs were defined as CD11b-CD11c+SiglecF+.

**Gene expression analysis on sorted airway AMs using NanoString nCounter system.** Total RNA was isolated from fixed, FACS-sorted airway-resident alveolar macrophages using the RNeasy FFPE kit (Qiagen) and the RNeasy Plus Mini Kit (Qiagen), following a modified version of the manufacturers' protocols[75]. Cells were first suspended in 240 μl of Buffer PKD. 10 μl of proteinase K was added and samples were incubated at 56 °C for 15 min, followed by a second incubation at 80 °C for 15 min. 500 μl of Buffer RBC was added and then samples were passed through the gDNA Eliminator column. 1200 μl of 100% ethanol was added to the flow-through and samples were passed through the RNeasy MiniElute spin column. Following washes with Buffers RW1 and RPE, samples were eluted with RNase-free water. RNA quality and concentration were determined using Thermo Scientific™ NanoDrop™ One Microvolume UV–Vis Spectrophotometer. All samples were found to have optimal RNA quality.

The nCounter® Mouse Inflammation V2 Panel CodeSet (NanoString Technologies, Seattle, WA, USA) profiling 254 genes: 248 mouse-inflammation genes and 6 housekeeping genes, was used to measure gene expression in our RNA samples from airway-resident AMs from Mtb-HT1 infected and Mtb-LT1 infected mice. Following the manufacturer's instructions, 150 ng of RNA from each sample was hybridized with the CodeSet for 18 h, and transcripts were quantitated using the nCounter® SPRINT Profiler (NanoString Technologies, Seattle, WA, USA) at Rutgers.

Raw data QC was performed using NanoString nSolver® version 4 software (NanoString Technologies, Seattle, WA, USA). Raw data were filtered to remove genes with poor binding and counts <100. The resulting 126 genes, 14 internal controls, and 6 housekeeping genes were used for subsequent normalization and analysis. Normalization of raw counts and differential gene expression analysis was performed using the NanoStringDiff (R package version 1.22.0, Bioconductor version release 3.13)[76], which utilizes a negative binomial-based generalized linear model to characterize raw data counts[77]. Data normalization parameters were estimated from positive controls, housekeeping genes, and background level obtained from negative controls. Normalized data were then used to calculate the log fold change (FC) of differentially expressed genes (DEG) between Mtb-HT1 and Mtb-LT1 infected airway-resident alveolar macrophages. DEG with $p < 0.05$ were identified and logFC values were graphed.

**Histopathological analysis of lung tissue.** Post-mortem, lungs of Mtb-infected mice were perfused with sterile PBS and fixed in 4% paraformaldehyde for 7 days, followed by paraffin embedding. For histopathological analysis, 5- to 7 μm sections were cut and stained using a standard H&E protocol. Brightfield 40X whole-slide scans were obtained using the Leica SCN400 F whole-slide scanner through collaboration with the Experimental Pathology Research Laboratory at NYU Langone Health. For subjective histological evaluations, qualitative metrics were recorded in a blinded fashion.

**Enzyme linked immunosorbent Assay (ELISA).** Lung lysates from the right superior lobe of Mtb infected mice were filtered through 0.22 μM Ultrafree centrifugal filters (Millipore UFC30GVNB) and treated with 2X Halt Protease inhibitor cocktail (Thermo Scientific 78439) and used as sample. IL-17A and IL-1β were detected using Ready Set Go ELISA kits (Invitrogen 88-7371-88 and Invitrogen 88-7013-88). IFNγ, IL-1β, IL-12p70 and TNF, concentrations were determined using V-PLEX Proinflammatory Panel Mouse Kit (MSD K15048D-2). MesoScale Discovery WORKBENCH analysis software 4.0.12 (LSR-4-0-12) was used to analyze MSD data.

**Adoptive transfer of CD3+ T cells.** C3HeB/FeJ mice were infected with either Mtb-HT1 or Mtb-LT1. 6–7 weeks post infection, lung draining lymph nodes and spleen tissue were harvested and single cells were prepared. CD3+ T cells were isolated by negative selection with the EasySep mouse T cell isolation kit (STEMCELL 19851). Isolated T cells from 5 mice per group were pooled and $2 \times 10^6$ T cells were retro orbitally injected to naïve mice. 72 h later, the recipient

mice were infected with 100–200 CFU of either Mtb-HT1, Mtb-LT1, Mtb-HT2, or Mtb-LT2. Appropriate tissues were harvested from the recipient mice at 11 weeks post infection.

**Statistical analyses and data visualization**. All statistical analyses were performed using Graph Pad Prism software. Outliers were identifies using the Grubb's method in Graph Pad Prism software. For the analysis of two groups, the unpaired t-test was used. For greater than two groups, one-way ANOVA with Tukey's correction was used. In all cases, $p$ value <0.05 was statistically significant. Graph Pad Prism software was used for all data visualization.

**Reporting summary**. Further information on research design is available in the Nature Research Reporting Summary linked to this article.

## Data availability

Below is the link to review BioProject: PRJNA759569. BioProject's metadata is available at https://www.ncbi.nlm.nih.gov/bioproject/PRJNA759569. H37Rv was the reference genome (GenBank accession number NC_000962.3). Our computational workflow for Mtb sublineage assignment is publicly available as part of the DuffyNGS R package https://github.com/robertdouglasmorrison/DuffyNGS. Additional data supporting the findings of this study are available in Supplementary Information. Source Data is provided as a Source Data file. Source data are provided with this paper.

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

## Acknowledgements
The work was funded by the National Institute of Allergy and Infectious Diseases, National Institutes of Health grants U19AI111276 and U01AI065663 to R.R.R., R.D., J.J.E. and P.S., and NIAID training grant T32AI125185 to A.L. The study sponsors were not involved in the study design, in the collection, analysis, and interpretation of data; in the writing of the manuscript; or in the decision to submit the manuscript for publication.

## Author contributions
P.S. and A.L. conceptualized the study; A.L., S.V., V.K., S.H., S.M., R.D.M., and D.R.S. contributed to acquisition and analysis of data; P.S., A.L., S.V., R.R.R., and J.J.E. contributed to data interpretation; P.S., J.J.E., R.R.R., and R.D. acquired the funding for the study; R.R.R., M.P., and R.D. were involved in setting up the study and in Mtb isolate collection; A.L. and P.S. wrote the original draft; P.S., A.L., S.V., V.K., J.J.E., R.R.R., S.M., R.D.M., D.R.S., M.P., and R.D. reviewed and edited the manuscript.

## Competing interests
The authors declare no competing interests.
