## [Peer Review File · Nature Communications]

Early alveolar macrophage response and IL-1R-dependent T cell Priming determine transmissibility of Mycobacterium tuberculosis StrainsREVIEWER COMMENTS

Reviewer #1 (Remarks to the Author):

This is a very interesting and well written manuscript describing the early differential immune responses to two Lineage 4 strains characterised by high and low transmissibility. The experiments are well performed and clearly described. The manuscript, despite its complexity, is easy to follow. The results support the conclusions. The figures are well presented, apart from the fact that the legend indicating Mtb-LT and Mtb-HT is missing after Figure 1, although the colour coding helps.

I only have a minor comment: the discussion could potentially be shortened and the authors should discuss what are the implications of their findings in TB control, treatment and intervention strategies. Page numbers are missing.

Reviewer #2 (Remarks to the Author):

In this manuscript, the authors extend their previous studies on the characterization of the early events following Mtb infection by using two Mtb strains: an Mtb strain associated with high transmissibility (HT) and an Mtb strain associated with lower transmissibility. Using the mouse model of infection, the authors elegantly highlight significant differences in the immunological mechanisms associated with the infections with the two strains and convincingly show that IL1R-dependent alveolar macrophages migration from the alveolar space is a critical step during Mtb HT infection that induces Th1-type immune response that is in turn instrumental for the induction of granulomas. Conversely, this mechanism is impaired during Mtb LT infection, where a higher bacterial load is achieved in the lung, with a Th-17 type immune response characterized by neutrophilic inflammation and higher bacterial burden.

Overall, the rationale of the study is sound, methodologies are appropriate, results well presented and conclusion appropriate.

The results obtained provide significant insights on the early events shaping Mtb infection.

It would be very useful to provide data on the genetic characterization of the two Mtb strains (HT and LT), so to pinpoint or at least provide some clues on the potential genetic features responsible for such remarkable differences in the in vivo data.

Likewise, it would be useful to investigate the mechanisms leading to the differential ability of the two strains to induce AM migration. Do the two strains show different ability to activate innate immune responses? Is it possible to address this issue by implementing relevant in vitro models of infection?

.

Response to Reviewer Comments

We are pleased with the overall positive comments by the reviewers. Below we provide a point-by-point response to the reviewer comments.

Reviewer #1

1. The legend indicating Mtb-LT and Mtb-HT is missing after Figure 1, although the colour coding helps.
We apologize for this oversight. This has been corrected in the revised manuscript.
2. The discussion could potentially be shortened and the authors should discuss what are the implications of their findings in TB control, treatment and intervention strategies.
This has been addressed.
3. Page numbers are missing.
This is now included in the revised manuscript.

Reviewer #2

1. It would be very useful to provide data on the genetic characterization of the two Mtb strains (HT and LT), so to pinpoint or at least provide some clues on the potential genetic features responsible for such remarkable differences in the in vivo data.
In Supplementary Table 1 we provide the WGS data of Mtb-HT1 and Mtb-LT1. The data show that the two strains show large number of polymorphisms. In future studies, we plan to perform WGS on additional strains, as well as RNAseq to provide insights into the genetic features regulating the in vivo phenotype.
2. Likewise, it would be useful to investigate the mechanisms leading to the differential ability of the two strains to induce AM migration. Do the two strains show different ability to activate innate immune responses? Is it possible to address this issue by implementing relevant in vitro models of infection?
This is addressed in Figure 6A where we provide new data from gene expression analysis of airway AMs isolated at day 13 from mice infected with Mtb-HT1 and Mtb-LT1 strains. We found that at this early timepoint following infection, AMs from Mtb-HT1-infected mice were preferentially activated to express several cytokines and chemokines, including IL-1 α and IL-1 β . A similar activation was not observed in AMs from Mtb-LT1 infected mice. These data indicate that the two strains show different ability to activate the innate immune responses.

REVIEWER COMMENTS

Reviewer #2 (Remarks to the Author):

I made two main comments to the previous version of the manuscript. While the authors properly addressed my comment #2 (activation of AMs), the comment on the genetic features of the HT and LT strains has not been addressed.

I think that it is important to indicate the lineage and clade of the two strains under study and discuss this information in line with current knowledge on the impact of Mtb genetic variability on TB pathogenesis.

Response to Reviewer Comments

We are pleased that, in principle, *Nature Communications* will publish a suitably revised version of our manuscript. Below we have addressed the comment from Reviewer 2.

Reviewer #2

I think that it is important to indicate the lineage and clade of the two strains under study and discuss this information in line with current knowledge on the impact of Mtb genetic variability on TB pathogenesis.

We analyzed whole genome sequencing data of Mtb-HT1 and Mtb-LT1 and our analysis pipeline assigned Mtb-HT1 strain to sublineage 4.1.2.1, with an edit distance of 1 (FDR = 0.0009); and assigned Mtb-LT1 strain to sublineage 4.3.4.2, with an edit distance of 2 (FDR = 0.0014). We have discussed this information in line with current knowledge on the impact of Mtb genetic variability on TB pathogenesis.